# A Novel Approach for Glioblastoma Treatment by Combining Apoptosis Inducers (TMZ, MTX, and Cytarabine) with E.V.A. (Eltanexor, Venetoclax, and A1210477) Inhibiting XPO1, Bcl-2, and Mcl-1

**DOI:** 10.3390/cells13070632

**Published:** 2024-04-04

**Authors:** Kai Zhao, Madita Braun, Leonie Meyer, Katharina Otte, Hartmann Raifer, Frederik Helmprobst, Vincent Möschl, Axel Pagenstecher, Hans Urban, Michael W. Ronellenfitsch, Joachim P. Steinbach, Jelena Pesek, Bernhard Watzer, Wolfgang A. Nockher, R. Verena Taudte, Andreas Neubauer, Christopher Nimsky, Jörg W. Bartsch, Tillmann Rusch

**Affiliations:** 1Department of Neurosurgery, Philipps University Marburg, Baldingerstraße 1, 35043 Marburg, Germany; 2Department of Hematology, Oncology & Immunology, Philipps University Marburg, Baldingerstraße 1, 35043 Marburg, Germany; 3FACS Core Facility, Philipps University Marburg, Hans-Meerwein-Straße 3, 35043 Marburg, Germany; 4Department of Neuropathology, Philipps University Marburg, Baldingerstraße 1, 35043 Marburg, Germany; 5University Cancer Center (UCT) Frankfurt—Marburg, Theodor-Stern-Kai 7, 60590 Frankfurt am Main, Germany; 6Dr. Senckenberg Institute of Neurooncology, Goethe-University of Frankfurt, Schleusenweg 2-16, 60528 Frankfurt am Main, Germany; 7Medical Mass Spectrometry Core Facility, Philipps University Marburg, Baldingerstraße 1, 35043 Marburg, Germany

**Keywords:** glioblastoma, temozolomide, TMZ, methotrexate, MTX, cytarabine, Ara-C, eltanexor, SINE, venetoclax, XPO1, Bcl-2, Mcl-1

## Abstract

Adjuvant treatment for Glioblastoma Grade 4 with Temozolomide (TMZ) inevitably fails due to therapeutic resistance, necessitating new approaches. Apoptosis induction in GB cells is inefficient, due to an excess of anti-apoptotic XPO1/Bcl-2-family proteins. We assessed TMZ, Methotrexate (MTX), and Cytarabine (Ara-C) (apoptosis inducers) combined with XPO1/Bcl-2/Mcl-1-inhibitors (apoptosis rescue) in GB cell lines and primary GB stem-like cells (GSCs). Using CellTiter-Glo^®^ and Caspase-3 activity assays, we generated dose–response curves and analyzed the gene and protein regulation of anti-apoptotic proteins via PCR and Western blots. Optimal drug combinations were examined for their impact on the cell cycle and apoptosis induction via FACS analysis, paralleled by the assessment of potential toxicity in healthy mouse brain slices. Ara-C and MTX proved to be 150- to 10,000-fold more potent in inducing apoptosis than TMZ. In response to inhibitors Eltanexor (XPO1; E), Venetoclax (Bcl-2; V), and A1210477 (Mcl-1; A), genes encoding for the corresponding proteins were upregulated in a compensatory manner. TMZ, MTX, and Ara-C combined with E, V, and A evidenced highly lethal effects when combined. As no significant cell death induction in mouse brain slices was observed, we conclude that this drug combination is effective in vitro and expected to have low side effects in vivo.

## 1. Introduction

Glioblastoma Grade 4 (GB) is a highly lethal brain tumor, noted for being the most prevalent and aggressive among adults [1,2]. The overall survival (OS) is approximately 10 to 15 months, with a five-year survival rate under 5% [3,4,5]. This applies, despite aggressive treatment, when combining surgical tumor resection with consecutive radio-chemotherapy. However grim these statistics already seem, they are based on large cohort studies that tend to primarily include patients who are in good physical condition, amongst them, patients with IDH-mutated gliomas, associated with better outcome [6]. Based on the most recent WHO 2021 classification of brain tumors, which excluded IDH-mutant tumors from the GB diagnosis, contemporary data indicate even lower survival rates for GB patients (median age 64), with OS ranging from 8.4 to 9.2 months combined with a significantly diminished quality of life [7,8]. Despite explicit advancements in neurosurgery, radiotherapy, and innovative new treatment options such as tumor-treating fields (TTF), there has not been a substantial improvement in patient outcomes over the past two decades [7,8,9]. In light of the current literature, we propose two major reasons for this.

### 1.1. Resistance to Apoptosis

GB cells exhibit a profound capability for evading apoptosis. Challenged with apoptotic triggers like radiotherapy or antitumor drugs, leading to increased selection pressure, GB cells deploy a variety of anti-apoptotic proteins [10,11]. This conducts an intrinsic deregulation of apoptotic cell death pathways and activation of pro-survival mechanisms, highlighting the adaptive resistance and resilience of GB cells in therapeutic contexts [12] (Figure 1).

Introduced by Stupp et al. in 2005, TMZ serves as the current standard adjuvant therapy for GB patients [14]. The therapeutic efficacy of this small (194 Da), lipophilic molecule hinges on its DNA methylation capability [15]. TMZ preferentially methylates DNA at the N7 positions of guanine in guanine-rich regions (approx. 70%) but also affects N3 positions of adenine (approx. 9%) and O6 guanine residues (approx. 6%) [16,17,18]. Repeated attempts of the cell to repair these methylations ultimately lead to DNA damage in the form of single and double strand breaks and subsequently activate intrinsic apoptosis via, i.a., p53 signaling pathways (Figure 1(1)).

In search of alternative inductors of apoptosis, we identified MTX and Ara-C as suitable candidates. Both cytostatics are already in clinical use for intrathecal administration in patients with meningeosis carcinomatosa or leukaemia, as well as for primary central nervous system (CNS) lymphomas [19,20,21]. Therefore, their safety profile for direct CNS application has been well established. MTX is a competitive inhibitor of dihydrofolate reductase (DHFR), which catalyzes the conversion of dihydrofolate into the active form tetrahydrofolate (THF). THF is essential for the de novo synthesis of nucleic acids DNA and RNA [22]. Ara-C is a pyrimidine analog which, after conversion into the triphosphate form (ara-CTP), competes with cytidine triphosphate (CTP) for incorporation into the DNA-strand by DNA-polymerase during replication [23]. Therefore, both substances are ultimately introducing DNA damage, thereby also activating the intrinsic pathway of apoptosis (Figure 1(1)).

Physiologically, DNA damage induced by the above-mentioned drugs results in the stabilization and activation of the tumor suppressor protein p53 [24]. Subsequently, p53 binds to DNA, inducing cell cycle arrest and DNA repair mechanisms. If damage accumulates to be substantial, p53 triggers the activation and transcriptional upregulation of BH3-only members, which belong to the pro-apoptotic Bcl-2 protein family. These members include BH3 Interacting Domain Death Agonist (BID), BCL-2 Associated Agonist of Cell Death (BAD), BCL-2-like protein 11 (BIM), BCL-2 Inter-acting Killer (BIK), Phorbol-12-myristate-13-acetate-induced protein 1 (NOXA), and p53 Upregulated Modulator of Apoptosis (PUMA) (Figure 1(2)) [13,25]. Upon activation, these proteins, among other functions, inhibit active anti-apoptotic proteins such as Bcl-2 and Mcl-1 [26]. Furthermore, the activities of the BH3-only members trigger the conversion of the mitochondria-associated proteins BAX and BAK into pore-forming oligomers within the mitochondrial membranes, initiating Mitochondrial Outer Membrane Permeabilization (MOMP) and the release of cytochrome c (Figure 1(3)) [27,28]. Through the formation of apoptosomes, cytochrome c and Apaf-1 activate Caspase 9, which in turn induces Caspases 7 and 3, culminating in the execution of apoptosis [13].

GB tumors show upregulated expression for various anti-apoptotic proteins, including XPO1, Bcl-2, and Mcl-1, in order to circumvent the aforementioned self-destruction [29,30]. The nuclear export protein exportin 1 (XPO1) maintains cellular homeostasis in healthy cells by transporting numerous RNA species and over 200 proteins, including tumor suppressors, from the nucleus to the cytoplasm [31,32]. In GB cells, overexpressed XPO1 exports nuclear tumor suppressors like p53 into the cytoplasm, hindering their functionality and inhibiting the initiation of intrinsic apoptosis (Figure 1(4)) [33]. Our prior research demonstrated that Eltanexor, an FDA-fast-tracked-approved second-generation XPO1 inhibitor, effectively diminishes GB and GB stem-like cell viability via induction of apoptosis and enhances radiation sensitivity at nanomolar concentrations [34,35].

Bcl-2 and Mcl-1 belong to the anti-apoptotic Bcl-2 protein family and bind to the proapoptotic proteins BAX and BAK, thereby inhibiting their oligomerization, keeping healthy cells away from self-destruction. Cancerous cells utilize their action to block the release of cytochrome c and reactive oxygen species, subsequently impeding apoptosis (Figure 1(5)) [36,37]. Venetoclax is a highly selective Bcl-2 inhibitor with FDA approval for patients with chronic lymphatic leukaemia (CLL) with an established clinical safety profile [38]. Interestingly, Bcl-2 inhibition triggers Mcl-1 upregulation in GB cells in order to compensate for the tasks Bcl-2 would usually undertake to save the cell from apoptosis [39]. Currently, the combination of Bcl-2- and Mcl-1-inhibitors (e.g., A1210477) is being proposed and discussed for treatment of various cancers in preclinical and clinical trials, including GB [40,41,42]. Furthermore, synergistical links between p53 and Mcl-1 have already been described [43].

### 1.2. Accessibility

The blood–brain barrier (BBB) effectively restricts most cancer therapeutics from entering the normal brain [44]. TMZ, while not the most effective apoptosis inducer in GB cells, is utilized clinically due to its BBB permeability. In contrast to other cancers’ treatment regimens that have undergone remarkable progress due to the oncological research advances of the past twenty years, assimilation of new therapeutics for GB patients seems to be impeded by the selective permeability of the BBB [45,46,47]. The pervasive accumulation of radiographic contrast material in GB tumors, which is typically impermeable to the brain, led to the discussion that the BBB is consistently compromised in GB patients. However, overwhelming clinical evidence demonstrates that there are tumor regions with an intact BBB, and a cure for GB will only be possible if these regions of tumor are adequately treated [48]. Isolated case reports present a cerebrospinal fluid (CSF) to plasma ratio for Venetoclax of 1:1000, supporting this thesis [49,50].

Considering the abovementioned thoughts, we came to the following conclusion: If GB cells evade apoptosis via several anti-apoptotic proteins and are also able to compensate the therapeutic inhibition of single anti-apoptotic proteins via upregulation of others in an interlinked manner, we propose combining apoptosis inducers in the form of chemotherapy with apoptosis rescue molecules E, V, and A. Additionally, an alternative route of administration needs to be considered in order to bypass the BBB. The purpose of this work was to evaluate the efficacy and potential toxicity of combinations with chemotherapy and E, V, and A in GB cell lines, GSCs, and healthy mouse brain slice culture.

## 2. Materials and Methods

### 2.1. Cell Culture

U87 and U251 GB cell lines, sourced from ECACC, were cultured in DMEM medium plus 10% fetal bovine serum (FBS), 1% penicillin/streptomycin, 1% non-essential amino acids (NEAA), and 1% Sodium pyruvate, incubated at 37 °C in a humidified incubator with 5% CO_2_. GSCs were isolated from GB patients as previously described [51]. These cells were cultured in DMEM/F12 medium plus 2% B27, 1% amphotericin, 0.5% HEPES, 0.1% gentamicin, EGF and bFGF at 20 ng/mL, in 100 mm dishes at 37 °C in a humidified incubator with 5% CO_2_.

### 2.2. Drugs

MTX (25 mg/mL) and Ara-C (100 mg/mL) were purchased from the pharmacy of the Philipps University Hospital Marburg. TMZ (S1237) and Eltanexor (S8397) were purchased from Selleck Chemicals (Houston, TX, USA). Venetoclax (HY-15531), A1210477 (HY-12468), and Staurosporine (HY-15141) were purchased from MedChemExpress (Monmouth Junction, NJ, USA).

### 2.3. Cell Viability Assay

Cell viability was assessed using the CellTiter-Glo 3D assay (G7571, Promega, Walldorf, Germany). U87 and U251 cells were seeded at 4.0 × 10^3^/well (MTX, Ara-C) or 2.0 × 10^3^/well (TMZ), and GSCs at 1.0 × 10^4^/well and incubated overnight. After treatment with drugs at indicated concentrations, viability for GB cell lines was measured after 3 (MTX, Ara-C) or 5 days (TMZ) and GSCs after 10 days. Before measurement, 50 μL of CellTiter-Glo 3D reagent was added, shaken for 15 min, and incubated for 15 min at RT in darkness. Luminescence was detected using a FLUOstar OPTIMA Microplate Reader (Offenburg, Germany).

### 2.4. RNA Isolation and Real-Time RT-PCR

RNA isolation followed a previously described method [52]. After treating cells with the indicated drugs, RNA was isolated using QIAzol (79306, Qiagen, Hilden, Germany), selecting an OD 260/280 ratio of 1.8 to 2.1. Consequently, 2 μg of RNA was converted to cDNA using RNA to cDNA EcoDry Premix (Takara, Kyoto, Japan). The RT-PCR reaction system comprised 10 μL SYBR Green/Rox Master Mix (Hercules, CA, USA), 2 μL primers, 6 μL nuclease-free water, and 2 μL cDNA. Initial denaturation was set at 95 °C for 10 min, followed by 40 cycles of 95 °C for 15 s and 60 °C for 1 min. XS-13 served as an internal reference. Primers for Bcl-2, Mcl-1, and XPO1 were purchased from Qiagen (Hilden, Germany). Gene expression changes were quantified in relation to control using 2^−ΔΔCT^, with heatmaps displaying relative expression data.

### 2.5. Gene Expression Analysis and Survival Curve Analysis

Bcl-2, Mcl-1, and XPO1 gene expression data were analyzed in the “Expression analysis Box Plots” part of the GEPIA2 website (http://gepia2.cancer-pku.cn/#analysis, accessed on 12 January 2024) to obtain the Bcl-2, Mcl-1, and XPO1 expression difference between GB tumor tissues and the normal tissues of the GTEx (Genotype-Tissue Expression) database. Additionally, we used the ‘Survival Map’ and ‘Survival Analysis’ modules to obtain OS and disease-free survival (DFS) data for GB patients.

### 2.6. Protein Isolation and Western Blot Analysis

Protein isolation followed a previously described method [52]. After drug treatment, cells were washed 3 times with ice-cold PBS, and total protein was extracted via RIPA buffer including phenantrolin, protease, and phosphatase inhibitors (A32955 + A32957, Thermo Scientific, Waltham, MA, USA). Protein-lysates were then boiled in a sample reducing buffer (B0009, Invitrogen, Waltham, MA, USA) and Laemmli for 5 min. Consequently, 20 µg protein were separated via 12.5% SDS-PAGE and transferred to NC membranes (A29591442, GE Healthcare Life Science, Solingen, Germany), blocked with 5% non-fat milk (T145.3, Carl Roth GmbH + Co. KG, Karlsruhe, Germany) for 1 h at RT. Membranes were incubated overnight at 4 °C with the following primary antibodies: Mcl-1 (1:1000 in 5% BSA in TBST, 94296, Cell Signaling Technology, Leiden, The Netherlands), Bcl-2 (0.1 µg/mL in in 5% milk in TBST, R&D System, Minneapolis, MN, USA), XPO1 (1:1000 in 5% BSA in TBST, 46249, Cell Signaling Technology, Leiden, The Netherlands), and β-tubulin (1:1000 dilution in 5% milk in TBST, NB600-936, Novus Biologicals, Littleton, CO, USA). Following 3 TBST-buffer washes, membranes were incubated with secondary antibodies Donkey Anti-Mouse (HRP) (dilution: 1:4000 in 5% milk in TBST; ab97030, Abcam, Cambridge, UK) and Donkey Anti-Rabbit (HRP) (dilution: 1:4000 in 5% milk in TBST; ab97064, Abcam, Cambridge, UK) for 1 h at RT. Following another 3 TBST-buffer washes, detection was performed via ChemiDoc MP Imaging System (Bio-Rad Laboratories GmbH, Feldkirchen, Germany).

### 2.7. Apoptosis Assay

Apoptosis was assessed using Caspase-Glo^®^ 3/7 Assay for GB cell lines and Caspase-Glo^®^ 3/7 3D Assay (G8090 + G8981, Promega GmbH, Walldorf, Germany) for GSCs. U87 and U251 cells (1 × 10^4^/well) and GSCs (4 × 10^3^/well) were seeded overnight and consequently treated with drugs at indicated concentrations. Apoptosis in U87 and U251 cells was measured at 24 h and in GSCs at 48 h post-treatment. Before measurement, 20 μL of the respective reagent was added, shaken for 30 s, incubated for 1 h at RT in the dark, and luminescence was detected via FLUOstar OPTIMA Microplate Reader (Offenburg, Germany).

### 2.8. Flow Cytometry Analysis

Propidium iodide (PI) staining (P4170, Sigma, Dreieich, Germany) was used for cell cycle analysis on 1 × 10^6^ cells seeded in T25 flasks overnight, consecutively treated with indicated drugs or DMSO as control for 24 h. Consequently, cells were washed with ice-cold PBS and fixed in 80% ethanol overnight, then washed with PBS again and stained with PI buffer containing 0.1% Triton X-100 (T8787, Sigma, Dreieich, Germany) and 1 mg DNase-free RNase A (EN0531, Thermo Fisher Scientific, Waltham, MA, USA), incubated for 3 min in the dark. Apoptosis FACS staining was performed by eBioscienceTM, Annexin V Apoptosis Detection Kit APC (88-8007-72, Invitrogen, Waltham, MA, USA). Treatment groups for apoptosis staining matched those of PI staining. All steps were executed according to the manufacturer’s instructions. Cells were washed with cold PBS and 1× binding buffer, followed by addition of 5 µL Annexin V to 100 µL of binding buffer and incubation for 15 min in the dark. Then, cells were washed with 1× binding buffer and resuspended in 200 µL of binding buffer, after adding 5 µL PI, and incubated for 30 min at RT in the dark. Finally, cells were washed in binding buffer and then resuspended in 400 µL binding buffer for FACS measurement.

### 2.9. Brain Slice Culture

Organotypic cerebellar slices were prepared using the interface method, with minor modifications [53,54]. Six-well plates were prepared with 1 mL culture medium per well, containing 25% (*v*/*v*) BME (21010046, Gibco^®^, Thermo Fisher Scientific, Waltham, MA, USA), 25% (*v*/*v*) heat-inactivated horse serum (H1270, Sigma-Aldrich^®^, Merck KGaA, Darmstadt, Germany), 1.3% (*v*/*v*) glucose (40% (*w*/*v*) stock solution, 2357742, B. Braun, Melsungen, Germany), and 1% (*v*/*v*) GlutaMax™ Supplement (35050061, Gibco^®^, Thermo Fisher Scientific, Waltham, MA, USA) in MEM (21575022, Gibco^®^, Thermo Fisher Scientific, Waltham, MA, USA). Experiments were performed with 8–9-day-old mice; after anesthesia with isoflurane, brains were removed, and the cerebellum was cut into 400 µm slices (Mcllwain™ Tissue Chopper). Slices were then transferred to an ice-cold dissection medium MEM (21575022, Gibco^®^, Thermo Fisher Scientific, Waltham, MA, USA), separated under a microscope, and subsequently placed on membranes of Millicell^®^CMz transwells with 0.4 µM pores (PICM0RG50). The membranes, containing 3 slices each, were placed in the previously prepared 6-well plates and cultivated at 37 °C in a humidified incubator with 5% CO_2_. Culture medium was replaced the next day with exchanges of culture medium every other day. After 8 days, the brain slices were treated with the following concentrations of drugs for 48 h: TMZ 750 µM, MTX 55 nM, Ara-C 8.5 µM, Venetoclax 10 µM, A1210477 10 µM, and Eltanexor 1 µM. Staurosporine (50 µM) was applied for 12 h and used as a positive control. DMSO served as vehicle control.

### 2.10. Immunofluorescence Staining for Brain Slice Culture

Twenty µL of 0.1 mM propidium iodide (P4170, Sigma-Aldrich^®^, Merck KGaA, Darmstadt, Germany) was added to the slice medium for 5–8 h, then replaced with 4% (*v*/*v*) PFA overnight at 4 °C for fixation. Slices were then washed with TBS (15 min) and PBS (30 min) and placed in blocking solution (1% (*w*/*v*) BSA, 0.3% (*w*/*v*) Triton, 0.1% (*w*/*v*) NaN3 in PBS) overnight at 4 °C. Consequently, slices were incubated with primary active caspase-3 antibodies (AF835, R&D Systems™, Bio-Techne, Minneapolis, MN, USA; in blocking solution 1:250) for 3 days on a slow-moving shaker. After washing 4x with PBS (30 min each), slices were incubated with the secondary antibody DyLight^®^ 488 (ab96919, abcam^®^, Cambridge, UK; in blocking solution without NaN_3_ 1:1000) for 3 days at 4 °C in the dark. After repeated washes with PBS (4× for 30 min each), slices were counterstained with Hoechst 33,342 dye (Sigma-Aldrich^®^, Merck KGaA, Darmstadt, Germany; in PBS 1:10,000) for 30 min and mounted on microscope slices. Pictures were taken using a Leica SP8i confocal laser scanning microscope.

### 2.11. Quantification of Neuronal Apoptosis in Mouse Brain Slice Cultures

For image analysis of apoptosis stains, the QuPath software (v0.5.0) was used [55]. In the first step, aggregates of red staining (propidium iodide) and green staining (caspase 3) were detected across the whole cerebellar slices using the cell detection command. For the detection of red staining (propidium iodide) aggregates, the following parameters were used: “detection channel”: propidium iodide (red channel), “requested pixel size”: 0 µm, “background radius”: 8 µm, “median filter radius”: 0 µm, “sigma”: 1.5 µm, “minimum area”: 10 µm^2^, “maximum area”: 400 µm^2^, “threshold”: 5, “cell expansion”: 0 µm. For the detection of green staining (caspase 3) aggregates, the following parameters were used: “detection channel”: caspase 3 (green channel), “requested pixel size”: 0 µm, “background radius”: 8 µm, “median filter radius”: 0 µm, “sigma”: 3 µm, “minimum area”: 5 µm^2^, “maximum area”: 600 µm^2^, “threshold”: 5, “cell expansion”: 0 µm. The parameters “use opening by reconstruction”, “split by shape”, “include cell nucleus”, “smooth boundaries”, and “make measurements” were activated in both cases. In the second step, some of these detections were visually deemed to be propidium iodide positive nuclei or caspase 3 positive cells, respectively. Detections that could not be deemed as either of these alternatives were set to be ignored (see Appendix A with “Training PI” and “Training Caspase 3”). Based on this visual classification, a random trees (RTrees) object classifier was trained to automatically classify the detected red staining (propidium iodide) and green staining (caspase 3) aggregates as propidium iodide positive nuclei or caspase 3 positive cells or to ignore them. Examples taken from all slices were used to train the classifier. This classifier was then applied to all slices, and numbers of propidium iodide positive nuclei or caspase 3 positive cells across the whole cerebellar slices were counted. For normalization, the number of PI-positive nuclei or caspase 3-positive cells per area was calculated.

### 2.12. Sampling of Liquor and Blood Samples from Patients

CSF and blood samples were collected under medically indicated clinical routines by the medical staff at the University Hospital Marburg, Department of Hematology, Oncology & Immunology, following established Standard Operating Procedures (SOPs). Cell-free CSF and plasma were obtained from supernatants after centrifugation, aliquoted, and frozen in liquid nitrogen. Samples were then pseudonymized, cataloged according to local data protection protocols, and stored at −80 °C in a designated freezer with restricted access. Informed consent was obtained from all patients who were 18 years old or older. Samples were exclusively used for this study and destroyed post-study. Only patients who had been on Venetoclax for chronic lymphocytic leukemia or other medically indicated reasons, taking 100 to 200 mg daily, were included, with sample collection occurring 5–8 h post oral administration to align with Venetoclax’s Tmax (time to reach maximal plasma concentration) to Cmax (maximal plasma concentration) [56,57]. Approval was obtained from the ethics committee of department 20 of the Philipps University Marburg (file number 105/20).

### 2.13. Quantification of Venetoclax in Patient-Derived Material

One hundred µL of liquor was mixed with an equal volume of methanol, containing Navitoclax as internal standard (1 ng/mL). For serum, 100 µL were mixed with 900 µL methanol, containing 10 ng/mL Navitoclax. The samples were centrifuged, and the supernatant was measured by LC-MS. Analysis was done on an Agilent 1290 HPLC coupled to a QTOF 5600 mass spectrometer (AB Sciex). Samples were separated on a Zorbax SB-C18 column (2.1 × 50 mm; Agilent) using a gradient as follows: 0 min (70% solvent B: Methanol), 2.5 min (90% B), 3 min (90% B), 3.1 min (70% B), 5.5 (70% B). Solvent A was 5 mM ammonium acetate. The flow rate was 0.6 mL/min. Navitoclax and Venetoclax were detected in product ion scan mode (positive ionization) and by monitoring the corresponding MSMS fragment (974 -> 742 and 868 -> 321, respectively). For quantification, a respective 6-point calibration curve was used for each sample type (liquor: 2–50 ng mL^−1^ and serum 0.1–10 µg mL^−1^). Calibration samples were prepared by spiking the appropriate amount of Venetoclax in drug-free liquor and serum, respectively. Data analysis was performed using Analyst TF 1.7.1 and MultiQuant 3.0.2 (AB Sciex).

### 2.14. Statistical Analyses

All data were shown as the mean ± SD or SEM and analyzed using GraphPad Prism software, Version 9.0 (GraphPad Software Inc., San Diego, CA, USA). The results were considered as not significant (ns, *p* > 0.05), and *p*-value < 0.05 was considered statistically significant. The IC50 value was determined by a non-linear regression method using the least-square fit. Analysis of variance (ANOVA) test was performed for multicomponent comparisons with consecutive post hoc test (Tukey).

## 3. Results

### 3.1. Cytotoxicity of TMZ, MTX, and Ara-C in GB Cell Lines and GSCs

We assessed the cytotoxicity of TMZ, MTX, and Ara-C on GB cell lines U87 and U251 as well as GSCs using the CellTiter-Glo 3D assay, comparing drug-treated groups against vehicle controls over 3 (MTX, Ara-C) or 5 (TMZ) days for GB cell lines and 10 days for GSCs, with all data normalized to the vehicle control. As indicated in Table 1, the IC_50_ values of TMZ, MTX, and Ara-C were 671.3 × 10^3^ nM, 59.87 nM, and 4886 nM for U87 GB cells, 48.22 × 10^3^ nM, 30.56 nM, and 1748 nM for U251 GB cells, and 68.86 × 10^3^ nM, 123 nM, and 367.7 nM for patient-derived GB stem-like cells, respectively (Table 1).

TMZ exhibited IC_50_ values exceeding 40 μM, notably in U87 cells, indicating relative resistance. Conversely, IC_50_ values for Ara-C ranged between 300 and 5000 nM, while MTX demonstrated the highest sensitivity among GB cells with IC_50_ values below 150 nM. These findings suggest MTX and Ara-C potentially offer greater efficacy in diminishing GB cell viability than TMZ. Remarkably, U87 cells, with highest resistance to TMZ, were markedly more sensitive to MTX, with IC_50_ ratio disparities exceeding 10,000-fold (Figure 2A). Additionally, GSCs, a niche group implicated in therapeutic resistance and tumor recurrence through mechanisms like radioresistance and chemoresistance, displayed sensitivity to Ara-C, with an IC_50_ ratio of 1:180 compared to TMZ (Figure 2B).

### 3.2. Expression Levels of Bcl-2, Mcl-1, and XPO1 in GB Cells, GSCs, and GB Patients

To elucidate the elevated expression of anti-apoptotic Bcl-2 family proteins (Bcl-2, Mcl-1) and XPO1 in GB, we analyzed their mRNA levels in established GB cell lines U87 and U251 as well as in patient-derived GSCs via qPCR. GSCs demonstrated significantly higher mRNA levels of Bcl-2, Mcl-1, and XPO1 (Figure 3A–C), aligning with their stem-like characteristics, compared to the established U87 and U251 cell lines. Additionally, TCGA and GTEx database analysis indicated notably higher expression levels of Bcl-2, Mcl-1, and XPO1 mRNA (Figure 3D–F) in clinical GB tumors compared to normal brain tissue, particularly significant for Mcl-1 (*p* < 0.01).

To illuminate the impact of Bcl-2, Mcl-1, and XPO1 gene expression on GB prognosis, we analyzed their correlation with GB patient outcomes using the GEPIA 2 database (Figure 3G,H,L). Lower Mcl-1 expression showed a trend towards improved OS (Figure 3H; *p* = 0.19) and correlated significantly with disease-free survival (DFS) (Figure 3K; *p* = 0.03). Conversely, OS and DFS differences between high and low expression groups for Bcl-2 (Figure 3G,J) and XPO1 (Figure 3I,L) were not statistically significant, indicating Mcl-1 expression’s unique prognostic significance in GB patient outcomes.

### 3.3. Treatment with E., V., and A. Results in Alterations of Bcl-2, Mcl-1, XPO1 Gene Expression at Both the Transcriptional and Translational Levels

As was shown before, treatment with Eltanexor induces XPO1 gene expression in U87 and U251 cells, likely as a compensatory response [34]. Given that Mcl-1 confers resistance to BH3-mimetics targeting Bcl-2 [41], we examined Bcl-2, Mcl-1, and XPO1 expression changes following treatment with BH3-mimetics (A1210477, Venetoclax) and Eltanexor, individually or in combination, in both GB cell lines and GSCs. qPCR was performed to detect Bcl-2, Mcl-1, and XPO1 mRNA expression post-treatment with the individual inhibitors alone (Figure 4A–C) or combined (Figure 4D–F).

U87 and U251 cells were treated at indicated concentrations for 12h, while GSCs received treatment for 48h. Our findings indicate that Venetoclax increased Mcl-1 mRNA levels in U87 and U251 cells but not in GSCs, whereas Bcl-2 and XPO1 levels remained constant. Interestingly, Venetoclax significantly reduced Bcl-2, Mcl-1, and XPO1 levels in GSCs (Figure 4A). Typically, Mcl-1 is upregulated in response to Bcl-2 inhibition [39], as observed in U87 and U251 cells. However, in GSCs, it may overcome the usual compensatory increase in Mcl-1.

Upon treatment with Venetoclax, Bcl-2 mRNA levels remained constant in U87 cells, yet there was a significant decrease in Bcl-2 protein levels (Figure 5A), thereby facilitating a pro-apoptotic condition. Additionally, Venetoclax substantially elevated Mcl-1 protein expression (Figure 5A–C) in all treated cells, indicating that translational upregulation of Mcl-1 by Venetoclax may contribute to resistance to this agent [58].

Furthermore, the effects of A1210477 treatment on Bcl-2, Mcl-1, and XPO1 expression were investigated. A1210477 slightly decreased Mcl-1 mRNA levels (Figure 4B), yet it elevated Mcl-1 protein levels in U87, U251, and GSCs (Figure 5A–C). This rise in Mcl-1 protein, despite reduced mRNA levels, was previously linked to an increased protein half-life [59]. Additionally, significant Bcl-2 down-regulation and XPO1 up-regulation in GSCs (Figure 4B) suggest that Mcl-1 inhibition may result in Bcl-2 down-regulation in GSCs, concurrently leading to XPO1 accumulation at the transcriptional level. The impact of Eltanexor on Bcl-2, Mcl-1, and XPO1 gene expression was assessed, revealing significant elevation in XPO1 mRNA levels, while Bcl-2 and Mcl-1 mRNA showed minor increases across all cell types (Figure 4C). This aligns with our previous findings that treatment with Eltanexor may be counteracted by specific compensatory mechanisms [34]. Notably, Eltanexor upregulated Mcl-1 mRNA in GSCs but had no effect on its protein expression (Figure 5A–C). In U251 cells, despite Bcl-2 and Mcl-1 protein levels remaining constant, their mRNA was upregulated by Eltanexor Figure 4C and Figure 5B), indicating a potential association with decreased function of XPO1 protein.

Meanwhile, Bcl-2, Mcl-1, and XPO1 mRNA levels were evaluated following combined treatments with Venetoclax, A1210477, and Eltanexor in U87 (Figure 4D) and U251 cells (Figure 4E), as well as GSCs (Figure 4F). Eltanexor significantly upregulated XPO1 across all cell types compared to the Venetoclax and A1210477 combination, suggesting that Eltanexor may invoke compensatory mechanisms to mitigate its own effects, unaffected by Venetoclax and A1210477. In both the Venetoclax and A1210477 (V+A) and the Venetoclax, A1210477, and Eltanexor (V+A+E) groups, Bcl-2 expression was reduced in U87 and GSCs but not in U251, with the reduction amplified by Eltanexor, promoting a pro-apoptotic state. In GSCs, Venetoclax and A1210477 co-treatment further diminished Mcl-1 levels (Figure 4F), whereas in U87 and U251 cells, Mcl-1 increased under V+A treatment (Figure 4D,E), a trend most pronounced in the absence of Eltanexor.

### 3.4. Evaluation of Cell Viability upon Combinatorial Treatment with E., V., A. and Chemotherapeutic Drugs in U87 and U251 Cells, as Well as GSCs

Given the limited efficacy and durability of monotherapy in cancer treatment, we explored a novel combination therapy framework, incorporating induction of apoptosis via chemotherapeutics (TMZ, MTX, and Ara-C) and rescue of apoptosis via BH3-mimetics (Venetoclax, A1210477) and Eltanexor. This approach was applied on two established GB cell lines (U87 and U251) for 3 (MTX, Ara-C) or 5 (TMZ) days and GSCs for 10 days. Cell viability outcomes were measured using CellTiter-Glo 3D assay.

Without additional chemotherapy, Venetoclax and A1210477 exhibited synergistic effects in U87 and U251 cells, whereas individually, neither Venetoclax nor A1210477 showed cytotoxicity in any cell line, including GSCs, even with combined Venetoclax and A1210477 treatment (Figure 6A,E,I). To assess synergism, we examined combinations of Eltanexor with BH3-mimetics, discovering that Eltanexor, when combined with Venetoclax and A1210477, synergistically diminished cell viability in U87 and U251 GB cell lines, as well as GSCs (Figure 6A,E,I). Additionally, Eltanexor significantly enhanced the cytotoxicity of either Venetoclax or A1210477 in U87 cells compared to monotherapy (Figure 6A).

Subsequently, we examined whether co-treatment of GB cell lines as well as GSCs with chemotherapeutics plus E., V., and A. enhances their cytotoxic efficacy. TMZ (Figure 6B,F,J), MTX (Figure 6C,G,K), and Ara-C (Figure 6D,H,L), in combination with E., V., and A., were employed to assess cell viability with concentrations approximating the IC50-values for each chemotherapeutic drug. The combination with E., V., and A. augmented TMZ, MTX, and Ara-C-mediated suppression of cell proliferation in all cells, yet showed no enhanced effect on GSCs compared to chemotherapy drugs combined with V. and A. This suggests that while chemotherapeutic drugs with V. and A. synergistically decrease GSC viability, Eltanexor does not further promote a pro-apoptotic transition in GSCs (Figure 6J–L), as it does in U87 (Figure 6B–D) and U251 (Figure 6F–H).

Eltanexor significantly decreased viability in U87 and U251 cells when combined with chemotherapeutic agents and either Venetoclax, A1210477, or V.+A. (Figure 6B–D,F–H), suggesting U87 and U251 cells exhibit greater sensitivity to Eltanexor upon co-treatment with chemotherapeutic agents and BH3-mimetics compared to GSCs. Collectively, these findings demonstrate that V.+A. and, especially, E.+V.+A. combinations with chemotherapy are significantly more effective in reducing cell viability than chemotherapy alone.

### 3.5. Evaluation of Cell Apoptosis upon Combinatorial Treatment with E., V., A. and Chemotherapeutic Drugs in U87 and U251 Cells, as Well as GSCs

To investigate the combination of E., V., and A. alone, or in co-treatment with either TMZ, MTX, or Ara-C on its potential to induce apoptosis in GB cell lines and GSCs, we used a luminogenic caspase-3 substrate. Treatment with the indicated combinations for 24 h (U87 and U251) or 48h (GSCs) revealed enhanced relative caspase activity for analysis. Significantly enhanced caspase activity was observed for the triple combination of E., V., and A. compared to treatments with V. or A. alone in all cells (Figure 7A,E,I). While Eltanexor alone minimally triggers apoptosis across all cell types, combinations of E.+V. or E.+A. significantly enhance caspase activity beyond that observed with Venetoclax or A1210477 alone (Figure 7A,E,I). The V.+A. combination significantly promotes apoptosis, a response with an indication for amplification by Eltanexor addition in U87 and GSCs (*p* > 0.05). Given that BH3-mimetic-based combination therapies primarily promote intrinsic apoptosis [60], our findings suggest Eltanexor’s limited contribution to pro-apoptotic cell death in the V.+A. context. However, in E.+V.+A. combinations, Eltanexor’s anti-neoplastic effects may be responsible for the observed increase in cell death.

Next, we examined whether co-treatment of E., V., and A. with chemotherapeutic drugs boosts caspase activity. The combination of either TMZ, MTX, or Ara-C with V.+A. or E.+V.+A. significantly triggered apoptosis across all investigated cell types (Figure 7B–L). These combinations turned out to be most effective, concerning the induction of apoptosis, with the indication of chemo plus E.+V.+A. being slightly more effective than chemo plus V.+A. (significantly for combinations with Ara-C in U87 and U251; Figure 7D,H). Interestingly, combinations of chemo plus E.+V. or E.+A. showed increased apoptosis induction compared to treatments with Eltanexor, Venetoclax, or A1210477 alone; notably in U87 (Figure 7B–D) and U251 cells (Figure 7F–H), suggesting heightened caspase activity in GB cells. This effect was not seen in GSCs (Figure 7J–L). Our findings indicate that E.+V., E.+A., V.+A., or E.+V.+A. combined with chemotherapeutic drugs notably enhance caspase activity, leading to pro-apoptotic cell death in GB cells and GSCs.

### 3.6. Detection of Cell Cycle Arrest and Apoptosis by Flow Cytometry

Activation of apoptosis coincides with “S” phase arrest. To identify cell cycle arrest, PI staining and FACS analysis were conducted in U87 (Figure 8A) and U251 (Figure 8B) cells. Cells were treated with Venetoclax, A1210477, and Eltanexor alone or in combination with chemotherapeutic drugs (TMZ, MTX, and Ara-C) for 24 h.

Upon treatment with V.+A. or E.+V.+A., both U87 and U251 cells exhibited reduced S phase entry, with a notable arrest in the G1 phase after co-treatment with E.+V.+A. plus chemotherapy, hindering progression to the S phase.

Additionally, apoptosis was specifically assessed using an Annexin V/PI double staining assay in U87 (Figure 9A) and U251 (Figure 9B) cells treated with Venetoclax, A1210477, and Eltanexor, either alone or combined with chemotherapy drugs for 24 h. Treatment with Venetoclax and A1210477 individually induced minimal apoptosis, yet their combination significantly enhanced apoptotic cell death in U87 (Figure 9C) and U251 (Figure 9D) cells. Furthermore, V.+A. augmented apoptosis in U251 cells even more when combined with Eltanexor. The incorporation of chemotherapeutic agents with E.+V.+A. also led to markedly increased apoptosis in both U87 and U251 cells (Figure 9C,D).

### 3.7. Apoptosis and Dead Cells Detected by Immunofluorescence Staining in Brain Slice Culture

BH3-mimetic-based combination therapies predominantly enhance intrinsic apoptosis [60]. We evaluated the previously described E.+V.+A. regimen alongside chemotherapeutic agents TMZ, MTX, and Ara-C on cultured healthy brain slices of mice. PI and Caspase 3 fluorescence staining was performed with Staurosporin serving as a positive control. Treatments were applied across different groups, incorporating E.+V.+A. combined with each chemotherapeutic drug. Blue Hoechst staining identified cell nuclei, red fluorescence indicated PI, and green fluorescence marked Caspase 3 (Figure 10). Strong PI and Caspase 3 signals were noted in the Staurosporin positive control. Treatment groups showed that E.+V.+A. combined with chemotherapy did not significantly upregulate Caspase 3 expression compared to chemotherapy alone, and we observed a drastic disparity in signal intensity of PI and Caspase 3 staining compared to the positive control group (Figure 10). This indicates the low cytotoxicity of our treatment regimen in healthy brain tissue.

### 3.8. Distribution of Venetoclax in CSF and Plasma of Patients after Oral Administration

Despite these encouraging results, the problem of tissue distribution remains a major obstacle. In a pilot experiment with two patients that were treated with oral Venetoclax (100 vs. 200 mg daily) for indicated medical reasons, we determined the concentration of Venetoclax in CSF and plasma 5 to 8 h after ingestion (Table 2). The concentrations in serum and in CSF were determined via mass spectrometry.

## 4. Discussion

For almost two decades, TMZ has been the gold standard for adjuvant chemotherapy for GB patients. As a lipophilic small molecule of 194 Da molecular weight, TMZ can pass the BBB and is activated by intracellular breakdown to yield a reactive moiety with an easily transferable methyl group. These favorable pharmacokinetic properties most likely contribute to the survival benefit in GB patient cohorts in clinical trials, and therefore, TMZ is the established chemotherapy for GB in clinical guidelines. However, there are several limitations associated with the use of TMZ which are primarily due to the inevitable development of therapeutic resistance of GB patients to this treatment. TMZ itself is a potent mutagen, leading to hypermutations in the genome of treated cells, which is associated with recurrence and resistance via malignant transformation [61]. TMZ is thought to induce apoptosis in GB cells. However, senescence of GB cells was also observed, making the cellular reaction to chemotherapy more complex. It is currently under debate if inefficient apoptosis caused by TMZ can cause release of death-associated molecular patterns (DAMPs), thereby provoking unwanted immune responses that counteract the therapeutic efficacy [61,62].

In contrast to TMZ, other cytostatic drugs such as MTX and Ara-C are more effective in inducing classical apoptosis in target cells in vitro. As shown in our study, in terms of IC_50_ values and dosage, MTX and Ara-C, both FDA-approved drugs for intrathecal administration, are by orders of magnitude more effective for induction of cell death. Notably, these drugs have a 1000- to 10,000-fold higher potency than TMZ in GB cell lines and GSCs. MTX (454 Da, hydrophilic) and Ara-C (250 Da, hydrophilic) [63,64] are probably not ideal for systemic administration in GB patients because of their expected poor penetration of the BBB. Studies with metastatic breast cancer showed that patients tolerated i.v. administration of 3 g/m^2^ to 3.5g/m^2^ well and showed efficacy, claiming that MTX penetrates the blood–brain barrier if one administers such high doses, claiming that even higher doses of MTX could be administered [65,66]. Also, for Ara-C, studies with liposomal encapsulation of Ara-C demonstrated efficacy in patients with glioma and leptomeningeal metastasis [67]. Nonetheless, high systemic doses will lead to side effects that can be avoided by intrathecal administration which have an established safety profile for CNS metastasized tumor entities [19,20,21].

Although apoptosis is effectively induced by the mentioned chemotherapeutics, GB cells are mastering versatile mechanisms of anti-apoptosis to survive chemotherapy. This includes the overexpression of anti-apoptotic proteins, such as Bcl-2 and Mcl-1, and of exportin XPO1. Especially pluripotent GSCs with extraordinary high expression levels of antiapoptotic proteins are the cells to target, as these give rise to recurrent tumors. A closer look at the features of GSCs revealed that in GB tumors, Mcl-1 is highly upregulated and seems to play an even more prominent role than Bcl-2, as demonstrated by the clinical data related to overall and progression-free survival. In addition to the observed overexpression of anti-apoptotic proteins, treatment with these inhibitor classes reveals compensatory counter-regulations of individual genes/proteins.

We can confirm previous findings of compensatory Mcl-1 upregulation after treatment with Venetoclax (Bcl-2 inhibitor) for GB cell lines. In contrast, Mcl-1 up-regulation in GSCs after Venetoclax treatment is less pronounced as these cells already show high Mcl-1 expression levels. Treatment with Mcl-1 inhibitor A1210477 leads to suppression of Mcl-1 expression, suggesting a regulatory autofeedback loop. A similar observation was made for Bcl-2 after treatment with Venetoclax in all GB cell lines and GSCs. When screening for essential anti-apoptotic proteins, we systematically investigated the Bcl-2 family. Upon inhibition of Bcl-2, Mcl-1 is upregulated, suggesting that a combination of V+A is obligatory to suppress a compensatory response. First clinical studies with AT101, a Bcl-2 family drug targeting Bcl-2, Mcl-1, Bcl-xL, and Bcl-w, in patients with initial and recurrent GB showed no real improvement in therapy with median survival rates similar to TMZ but low toxicity [68]. We conclude that even if anti-apoptotic mechanisms are effectively suppressed, i.e., by treatment with the most specific Mcl-1 inhibitor, GSC cells show a compensatory upregulation of XPO1. This clearly indicates a crosstalk between the mitochondrial and the nuclear control of apoptosis, thereby safeguarding a fine-tuning of anti-apoptosis pathways in GB cells. As transcriptional data demonstrate, this type of crosstalk works in both directions, as Mcl-1 is upregulated in established cell lines under conditions of XPO1 inhibition. Bcl-2 and Mcl-1 are suppressed upon therapy with Bcl-2/Mcl-1 and XPO1 inhibition. On the protein level, this observation can be partially confirmed. For instance, GSCs respond to treatment using Bcl-2 and Mcl-1 inhibitors with downregulation of Bcl-2 and Mcl-1, respectively, whereas XPO1 is upregulated as a consequence of combined treatment with XPO1 inhibitors.

Together, these findings support the notion that only a combination of all three inhibitors is effective in preventing a proper cellular anti-apoptotic response which eventually causes tumor chemoresistance. The XPO1 inhibitor Eltanexor stimulates up-regulation of XPO1 as a compensatory mechanism in all GB cells investigated. Further investigation of the cell death mechanism determining Caspase 3 activities suggests that the combined treatment of cells with low doses of inhibitors E.,V., and A. amplifies the apoptosis rate of chemotherapy drugs such as TMZ by at least 4-fold, suggesting that the therapeutic limitations resulting from single chemotherapy treatment can be overcome by combined therapy with Venetoclax, A1210477, and Eltanexor. Compared to TMZ, much higher apoptosis rates were observed for MTX and Ara-C, giving rise to the notion that these drugs might be more suitable for treatment of GB if an efficient application route can be provided. As demonstrated in mouse brain slice cultures, neural apoptosis rates detected with an antibody directed against activated caspase 3 were comparably low for most combinations with the only exception of MTX with E., V., and A., which was only slightly enhanced. When apoptosis inducers were compared, PI staining revealed a slightly lower rate of cell death in TMZ-treated slices compared to the ones treated with MTX and Ara-C. However, this does not rule out the use of these drugs in future combinations in clinical studies. To establish these treatments as therapy, additional analyses of systemic toxicity in tissues/organs with high proliferation rates, such as bone marrow, are required.

With regard to systemic application routes, various studies have reported that in humans, the plasma levels of Venetoclax range between 30 and 4000 ng/mL, depending on the dosage administered (from 100 to 600 mg) [69]. In the quest to analyze whether oral Venetoclax administration leads to sufficient doses in the CNS, we used CSF as an indicator for distribution. Oral application of Venetoclax, as exemplified for two patients, resulted in approximately 1/340th of the serum concentration in cerebrospinal fluid and is hence not efficient to reach therapeutic doses in patients with an intact blood–brain barrier. Although it is being discussed whether the BBB is consistently compromised in GB patients, overwhelming clinical evidence demonstrates that there are tumor regions with an intact BBB, and a cure for GB will only be possible if these regions of tumor are adequately treated [48]. Isolated case reports present a cerebrospinal fluid (CSF) to plasma ratio for Venetoclax of 1:1000, supporting this thesis [49,50]. Alternative routes of administration, such as direct application of drugs to the resection cavity via specific delivery routes (e.g., Ommaya reservoir), need to be evaluated.

Although we demonstrated significant results for the selected drugs, several aspects need to be considered as limitations of our study: (i) with regard to translation into the clinic, several other inhibitors for XPO1 and Bcl-2 family pathways are already in use, such as MIK665 for Mcl-1, Selinexor for XPO1, and Navitoclax for Bcl-2 family proteins; (ii) it remains to be clarified whether other modes of cell death are of importance for the observed effects when using these drug combinations; (iii) although the mouse brain slices are informative for neuronal/astrocytic cell death using the drug combinations, higher neuronal functions, reflecting on behavior, for example, cannot be assessed, which requires further animal studies to analyze drug tolerability.

## 5. Conclusions

Combining a chemotherapeutic drug (TMZ, MTX, Ara-C) with inhibitors of Bcl-2, Mcl-1, and XPO1 in individually non-toxic doses is sufficient to induce highly efficient apoptosis of GB cell lines and stem-like cells in vitro with all relevant features whilst keeping in vivo neural toxicity with regard to cell death induction low. Our data suggest that such a combination, suitable application routes provided, could have therapeutic benefit for GB patients and might outperform therapy with TMZ.

## Figures and Tables

**Figure 1 cells-13-00632-f001:**
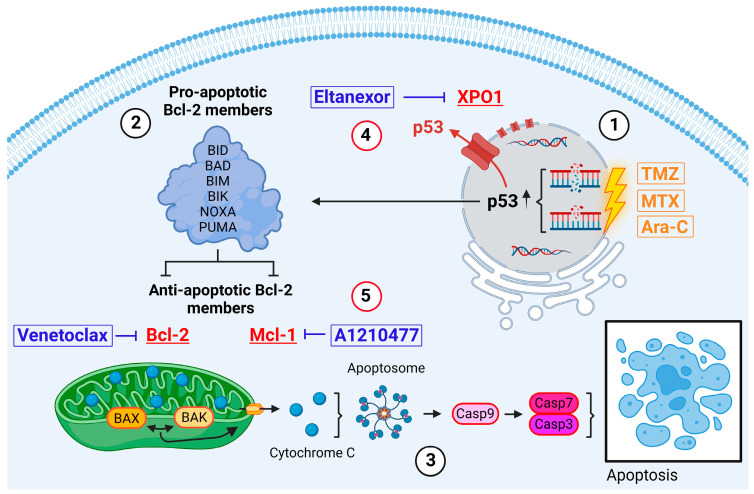
Intrinsic pathway of apoptosis exemplified in GB cells. Induced via, i.a., intrinsic stress, DNA damage, hypoxia (direct induction via TMZ, MTX, or Ara-C) [13]. Created with BioRender.com.

**Figure 2 cells-13-00632-f002:**
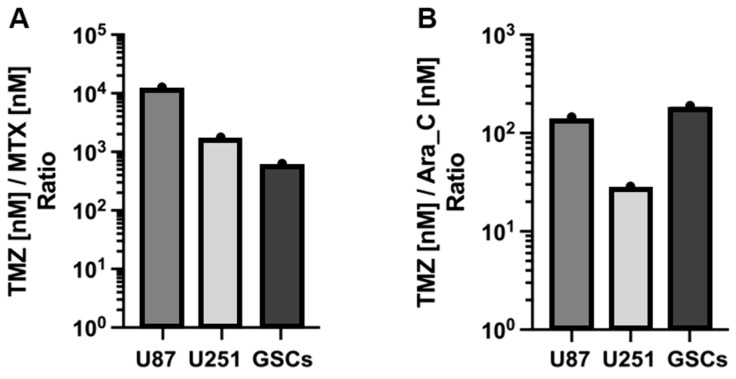
Histogram of the ratio of TMZ to MTX (**A**) and to Ara-C (**B**) based on IC_50_ values in Table 1, respectively. All cells are significantly more resistant to TMZ in comparison to MTX and Ara-C. E.g.: MTX:TMZ ≙ approx. 1:10,000 in U87; Ara-C:TMZ ≙ approx. 1:180 in GSCs.

**Figure 3 cells-13-00632-f003:**
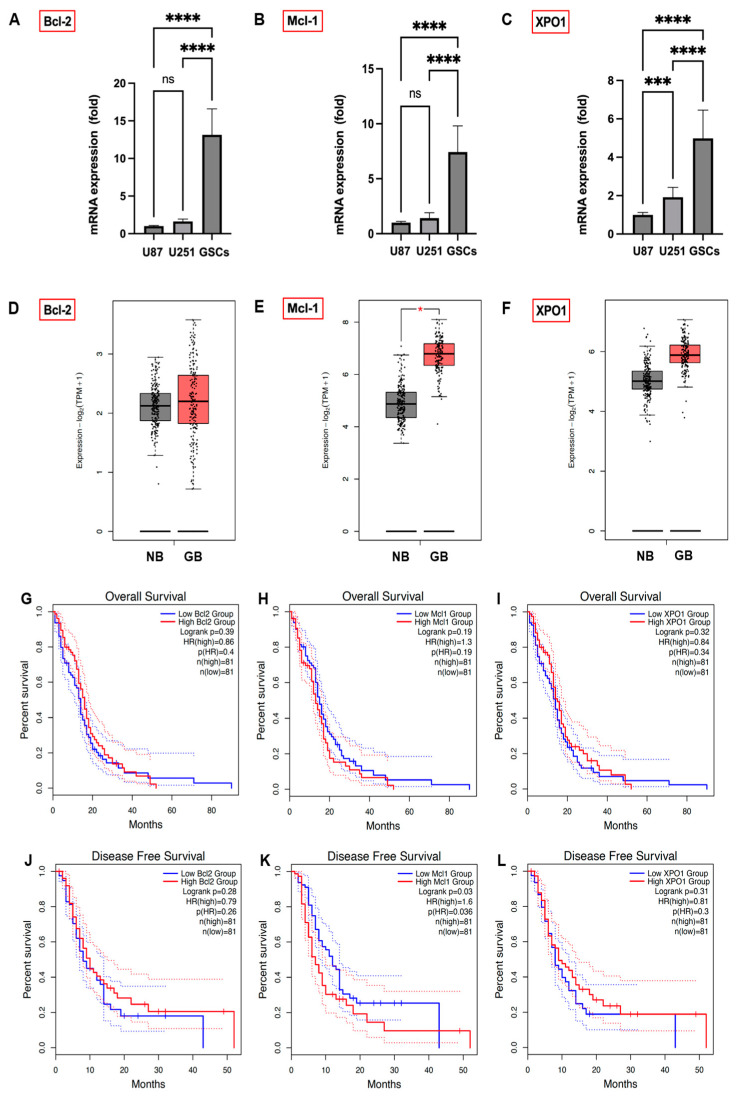
mRNA expression of Bcl-2 (**A**), Mcl-1 (**B**), and XPO1 (**C**) in GB cell lines U87 and U251 and patient-derived GSCs, quantified by qPCR, with U87 cell expression normalized to 1. Data were acquired from three independent experiments performed in triplicates and are presented as mean ± SD. One-way ANOVA with consecutive post hoc test (Tukey) was used for analysis; *** *p* < 0.001, **** *p* < 0.0001, ns: not significant. The association between Bcl-2, Mcl-1, and XPO1 gene expression in GB patients and their prognostic outcomes. The expression status of Bcl-2 (**D**), Mcl-1 ((**E**); * *p* < 0.05), and XPO1 (**F**) in clinical GB tumor tissue compared to normal brains was analyzed from the TCGA and GTEx databases. The GEPIA 2 database was used to analyze OS and DFS with a median group cutoff for either high (red) or low (blue) gene expression of Bcl-2 (**G**,**J**), Mcl-1 (**H**,**K**) and XPO1 (**I**,**L**) in GB patients.

**Figure 4 cells-13-00632-f004:**
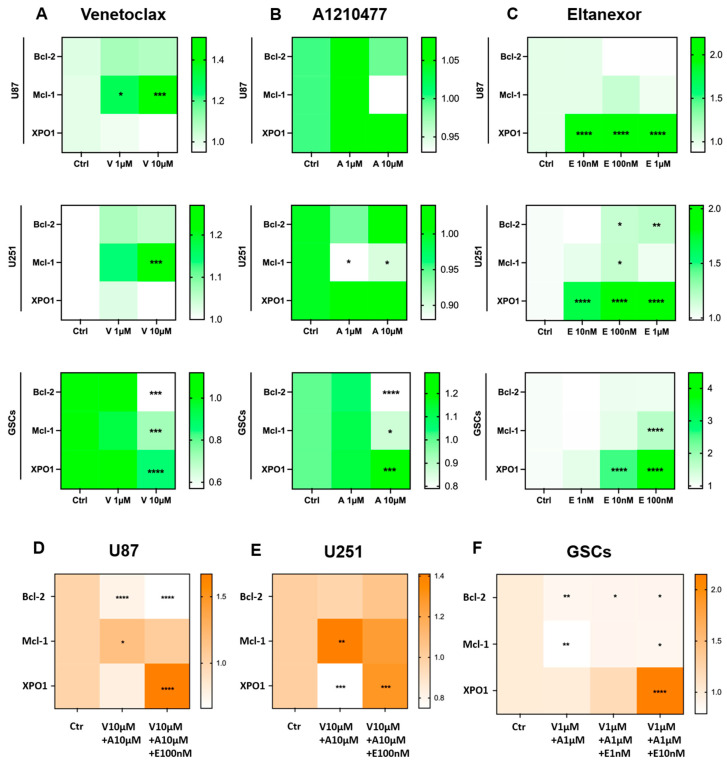
Modifications in *Bcl-2*, *Mcl-1*, and *XPO1* gene expression at the transcriptional level after treatment with Venetoclax (**A**), A1210477 (**B**), and Eltanexor (**C**), alone or in combination (**D**–**F**) in U87 and U251 cell-lines (12h of treatment) and GSCs (48h of treatment). Data are based on 3 independent experiments, utilizing qPCR for quantification. The heatmaps were generated from mean expression data. * *p* < 0.05; ** *p* < 0.01; *** *p* < 0.001; **** *p* < 0.001.

**Figure 5 cells-13-00632-f005:**
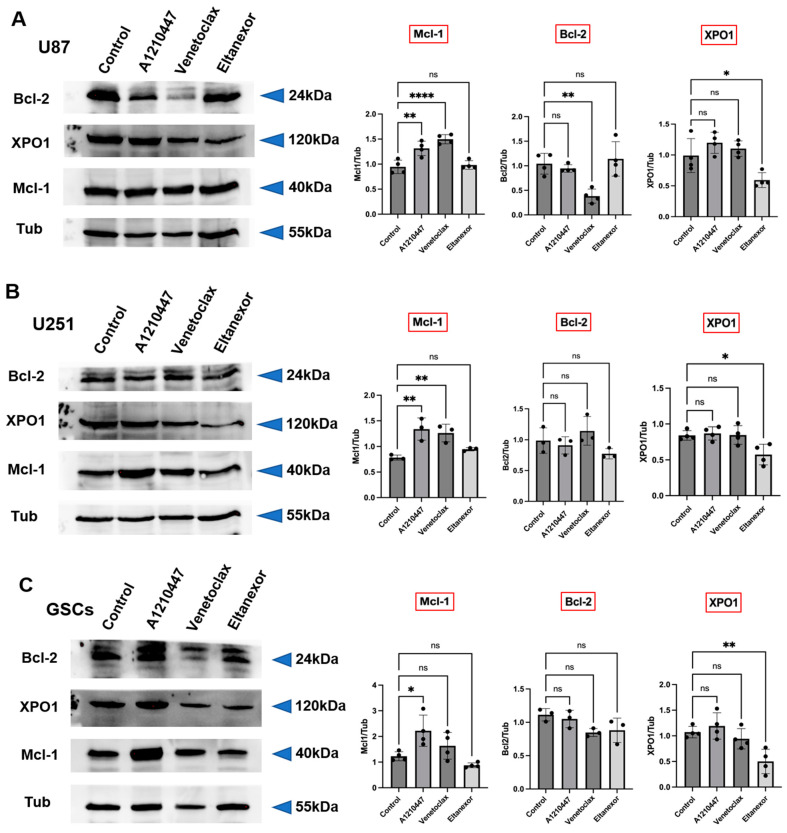
Modifications in Bcl-2, Mcl-1, and XPO1 protein level after treatment with Venetoclax (10 µM), A1210477 (10 µM), and Eltanexor (100 nM) in U87 (**A**) and U251 (**B**) cells, as well as GSCs (**C**). Western blot analyses depict the protein induction post-treatment with indicated drugs. Quantitative assessments of blots were obtained from three to four independent experiments relative to the control group. Data are presented as mean ± SD. One-way ANOVA with consecutive post hoc test (Tukey) was performed for statistical evaluation. * *p* < 0.05; ** *p* < 0.01; **** *p* < 0.001, ns: not significant.

**Figure 6 cells-13-00632-f006:**
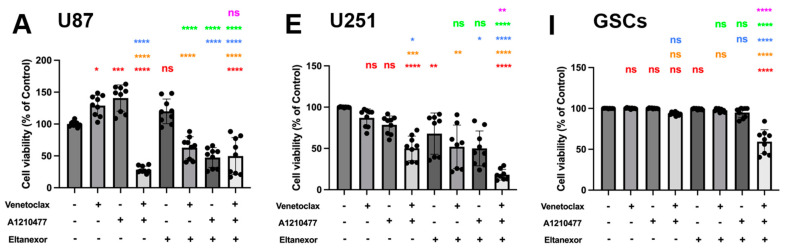
Cell viability of combinatorial treatment with TMZ, MTX, and Ara-C and E., V., and A., assessed via CellTiter-Glo 3D. The left panel represents the quantification of cell viability for U87 cells (**A**–**D**); (TMZ 750 µM, MTX 55 nM, Ara-C 8.5 µM). The middle panel (**E**–**H**) represents the quantification of cell viability for U251 cells (TMZ 100 µM, MTX 30 nM, Ara-C 2.5 µM). Both cell lines have been treated with drugs for 3 or 5 days, and the concentration of E., V., and A. were the same (V. 10 µM, A. 10 µM, E. 100 nM). The right panel (**I**–**L**) represents the quantification of cell viability for GSCs after 10 days of treatment (TMZ 20 µM, MTX 55 nM, Ara-C 500 nM., V. 1 µM, A. 1 µM, E. 10 nM,). Histograms are shown in relation to DMSO-control. (**A**,**E**,**I**) **Red**: comparison to control; **Orange**: comparison to V.; **Blue**: comparison to A.; **Green**: comparison to E.; **Purple**: comparison to V.+A. (**B**–**D**,**F**–**H**,**J**–**L**) **Red**: comparison to chemo-drug; **Orange**: comparison to chemo-drug+V.; **Blue**: comparison to chemo-drug+A.; **Green**: comparison to chemo-drug+E.; **Purple**: comparison to chemo-drug+V.+A. Results were obtained from three independent experiments performed in triplicates, and data are presented as mean ± SD. One-way ANOVA with consecutive post hoc test (Tukey) was used to analyze; * *p* < 0.05; ** *p* < 0.01; *** *p* < 0.001, **** *p* < 0.001, ns: not significant.

**Figure 7 cells-13-00632-f007:**
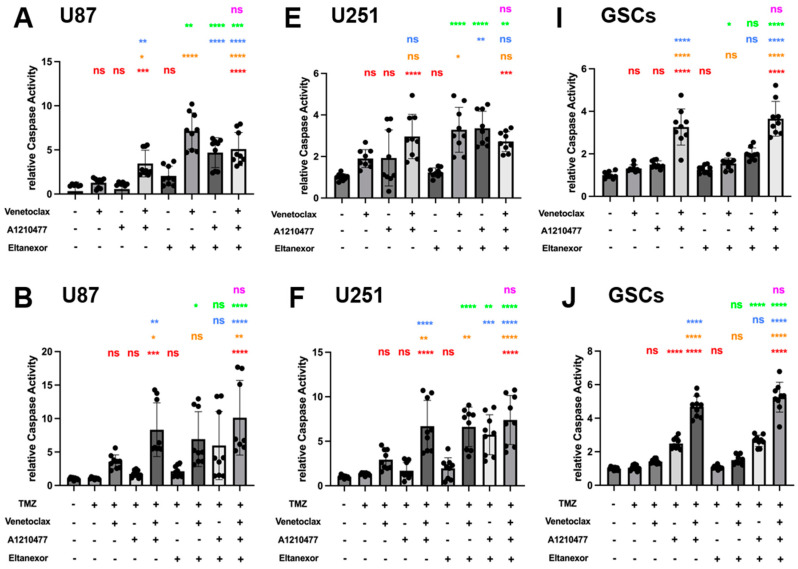
Chemo drugs (TMZ, MTX, and Ara-C) combined with E., V., and A. induced apoptosis in U87 and U251 GB cell lines, as well as GSCs, evaluated via Caspase GLO assays for Caspase 3/7 activity. Treatments were applied for 24 h in U87 (**A**–**D**) and U251 cells (**E**–**H**), and 48h in GSCs (**I**–**L**), consistent with Figure 6. (**A**,**E**,**I**) **Red**: comparison to control; **Orange**: comparison to V.; **Blue**: comparison to A.; **Green**: comparison to E.; **Purple**: comparison to V.+A. (**B**–**D**,**F**–**H**,**J**–**L**) **Red**: comparison to chemo-drug; **Orange**: comparison to chemo-drug+V.; **Blue**: comparison to chemo-drug+A.; **Green**: comparison to chemo-drug+E.; **Purple**: comparison to chemo-drug+V.+A. Results were obtained from three independent experiments, data are presented as mean ± SD. One-way ANOVA with consecutive post hoc test (Tukey) was used to analyze; * *p* < 0.05; ** *p* < 0.01; *** *p* < 0.001, **** *p* < 0.001, ns: not significant.

**Figure 8 cells-13-00632-f008:**
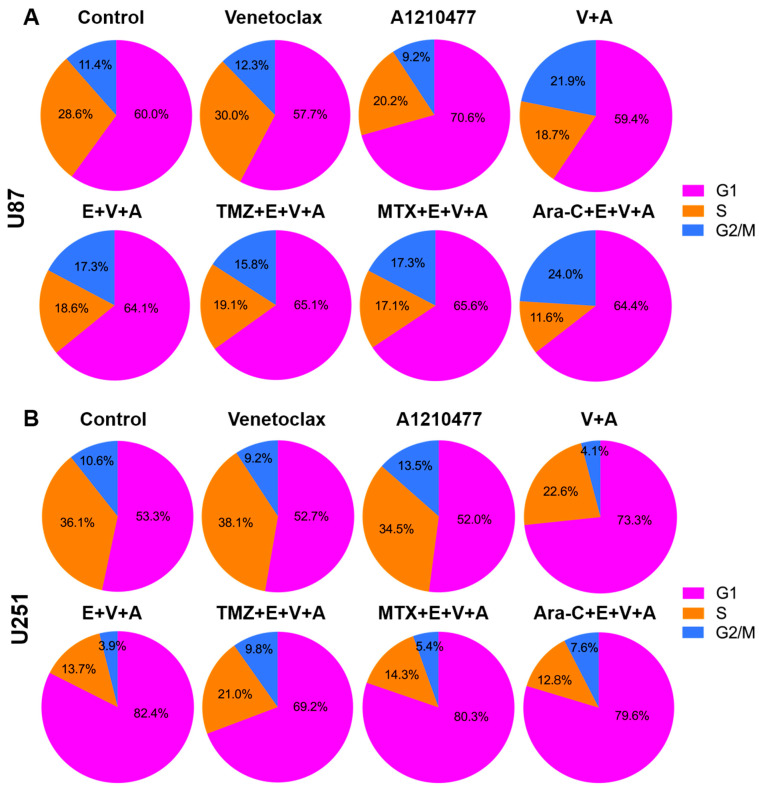
Cell cycle analysis was conducted using flow cytometry after propidium iodide staining, with cells treated as described in Figure 6 for 24 h. Pie charts depict the distribution across cell cycle phases (G1, S, G2) in U87 (**A**) and U251 (**B**) cells. Data shown are based on one representative experiment of three independent replications.

**Figure 9 cells-13-00632-f009:**
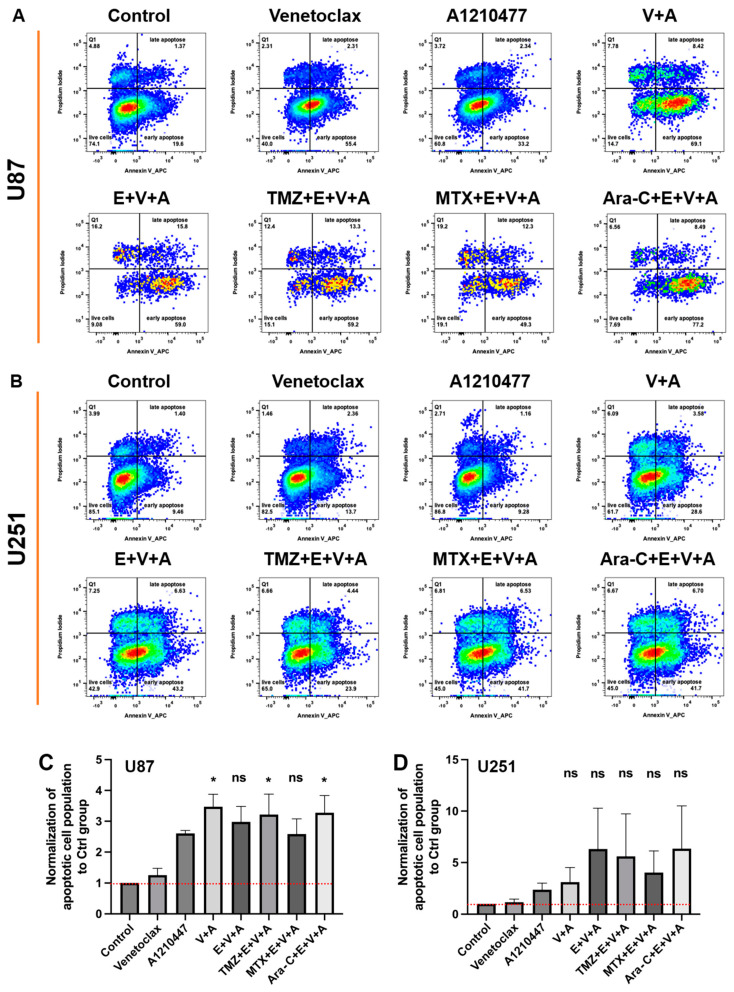
Apoptosis in U87 (**A**) and U251 (**B**) cells was assessed post-treatment with Venetoclax, A1210477 (individually as well as combined), and chemotherapeutic drugs TMZ, MTX, and Ara-C combined with E.+V.+A., using the same concentrations as in Figure 6 for 24 h, via FACS analysis of Annexin V stainings. Quantitative analysis of Annexin V staining from three independent experiments post-treatment in U87 (**C**) and U251 (**D**) cells is presented as mean ± SEM. One-way ANOVA with consecutive post hoc test (Tukey) was used to analyze the data; * *p* < 0.05; ns: not significant.

**Figure 10 cells-13-00632-f010:**
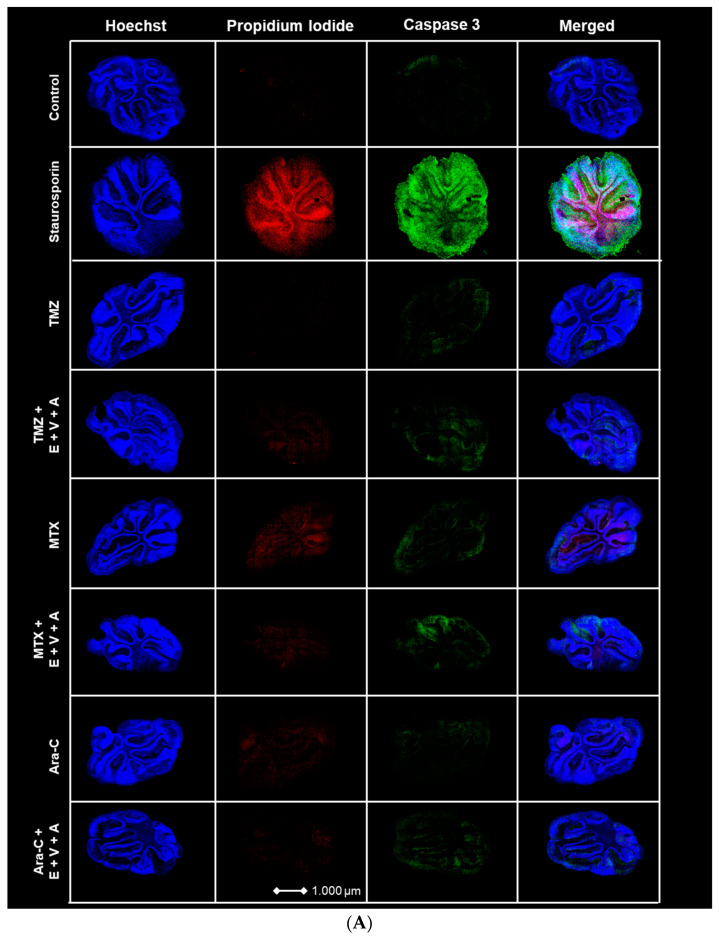
(**A**) Apoptosis and dead cell staining of brain slice culture with propidium iodide (dead cells; red), and Caspase 3 (apoptotic cells; green) post-treatment with either TMZ, MTX, and Ara-C and in combination with E.+V.+A. Staining with Hoechst indicates cell nuclei (blue). The left panel consists of merged images from Hoechst-, PI-, and Caspase 3 staining. Staurosporine was used as positive control for induction of apoptosis. (**B**) Quantification of morphological image data from Figure 10A. Dotted red line indicates the threshold of the control group. Staurosporine was used for positive control. Numbers of PI+ cells/mm^2^ are shown in logarithmic scale. One cerebellar slice per treatment was analyzed.

**Table 1 cells-13-00632-t001:** IC_50_ values for TMZ, MTX, and Ara-C for GB cell lines U87, U251, and patient-derived GSCs. All concentrations were determined by three independent experiments. Number of viable cells was determined by CellTiter-Glo reagent, and IC_50_ values were calculated using non-linear regression with the least-square fit.

Cells	IC_50_ (nM)		
	TMZ	MTX	Ara-C
**U87**	671.3 × 10^3^	59.87	4886
**U251**	48.22 × 10^3^	30.56	1748
**GSCs**	68.86 × 10^3^	123.0	367.7

**Table 2 cells-13-00632-t002:** Distribution of Venetoclax in serum and CSF after oral administration in two patients.

	Daily Dose (p.o.)	Serum	CSF	Ratio (⌀)
**Patient 1**	100 mg	230 ng/mL	$\left.\begin{array}{c} 0.5\;ng/mL\\ 9.9\;ng/mL \end{array}\right\}$	340:1
**Patient 2**	200 mg	2200 ng/mL		

## Data Availability

All data generated in this study are available on reasonable request.

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
