# Peer review of "A Novel Approach for Glioblastoma Treatment by Combining Apoptosis Inducers (TMZ, MTX, and Cytarabine) with E.V.A. (Eltanexor, Venetoclax, and A1210477) Inhibiting XPO1, Bcl-2, and Mcl-1"

_cells, 2024, doi:10.3390/cells13070632_

Round 1

Reviewer 1 Report

Comments and Suggestions for Authors

The authors investigated the combination of apoptosis inducers (TMZ, Methotrexate, and Cytarabine) with apoptosis rescue inhibitors (Eltanexor, Venetoclax, and A1210477) in GB cell lines and primary GB stem-like cells (GSCs). Using CellTiter-Glo® and Caspase-3 activity assays, they assessed dose-response curves and analyzed gene and protein regulation of anti-apoptotic proteins via PCR and Western Blots. Optimal drug combinations were evaluated for their impact on cell cycle and apoptosis induction using FACS analysis, along with potential toxicity in healthy mouse brain slices. Ara-C and MTX exhibited significantly greater potency in inducing apoptosis than TMZ. In response to the inhibitors Eltanexor (XPO-1; E), Venetoclax (Bcl-2; V), and A1210477 (Mcl-1; A), corresponding gene upregulation was observed in a compensatory manner. The combination of TMZ, MTX, and Ara-C with E, V, and A showed highly lethal effects when combined. No significant cell death induction was observed in mouse brain slices, suggesting low in vivo toxicity. Minor corrections may be needed for publication.

Introduction:

  1. The introduction paints a vivid picture of glioblastoma Grade IV (GB), highlighting its devastating impact and aggressive nature among adults. However, let's swap out "Glioblastoma °IV" with "Glioblastoma Grade IV" to make it clearer for everyone.

In section 1.1

  1. When discussing how TMZ methylates DNA, it might be helpful to explain briefly how these methylation patterns lead to DNA damage and eventual cell death through apoptosis. ["TMZ preferentially methylates DNA at the N7 positions of guanine in guanine-rich regions (approx. 70%), but also affects N3 positions of adenine (approx. 9%) and O6 guanine residues (approx. 6%),"]
  2. While the paragraph on MTX and Ara-C inducing apoptosis is crystal clear, it would flow even better if it could smoothly link this discussion back to the earlier conversation about apoptosis resistance in GB cells.
  3. That sentence about Eltanexor's efficacy in diminishing GB cell viability is impressive! It would be great to add a quick note on why this finding is significant or how it contributes to our overall understanding of apoptosis resistance in GB. [Line 108-112: "Our prior research demonstrated that Eltanexor, an FDA fast-tracked approved second-generation XPO1 inhibitor, effectively diminishes GB and GB stem-like cell viability via induction of apoptosis and enhances radiation sensitivity at nanomolar concentrations,"]

In section 1.2

  1. The transition between discussing the blood-brain barrier and proposing a new treatment approach could use some smoothing out. Making this connection more explicit would help tie everything together seamlessly.
  2. While the sentence about the study's purpose is straightforward, it would be even clearer if it could mention the apoptosis inducers E, V, and A again to provide context for the reader.

Materials and Methods:

1.    Typo noticed in lines 152 and 246.

Results:

2.    Discrepancies in figure numbering were noted. It is considered good practice to name figures sequentially, rather than using both Roman numerals and Arabic numerals.

3.    It appears that statistical analysis is missing for Fig. 8C and D. [Treatment with Venetoclax and A1210477 individually induced minimal apoptosis, yet (line 554) their combination significantly enhanced apoptotic cell death in U87 (Figure 8 C) and (line 555) U251 (Figure 8 D) cells.]

Discussion:

No comments. Overall, the discussion provides a thorough analysis of the experimental findings and their implications for GB treatment, while also identifying important avenues for future research and clinical investigation.

Conclusion:

No comments. Overall, the conclusions provide a concise summary of the study's findings and underscore the potential of combination therapies involving chemotherapeutic drugs and inhibitors of anti-apoptotic pathways as a promising approach for improving GB treatment outcomes.

Reviewer 2 Report

Comments and Suggestions for Authors

In the current study, the authors combined chemotherapeutic drugs TMZ, MTX, and Ara-C along with the inhibitors of the arm of signaling pathways that result in the resistance of these chemotherapeutic drugs due to apoptosis rescue. They obtained a significant downregulation of cell viability and induction in cell death with the combination therapy. The study is interesting and very well designed. I just have few concerns given below.

·      The authors have started the abstract introducing the delimitation of TMZ in GBM and in the next sentence have suggested a combination of drugs without giving any rationale. The connectivity is missing here. Please indicate why TMZ,MTX and Ara-C combined with XPO1/Bcl-2/Mcl-1-was used in GB cell lines in Abstract

·      Please write the interpretation of the results in the last paragraphs

·      Please indicate if it is SD or SEM in figure legends

·      Have you also performed IC50 for Venetoclax, A1210477, and Eltanexor.  Also cytotoxicity at this concentration in normal cells to exclude off-target effect?

·      Could you also show some in vivo experiments?

Reviewer 3 Report

Comments and Suggestions for Authors

The manuscript(cells-2905175) entitled “A novel approach for glioblastoma treatment by combining apoptosis inducers (TMZ, MTX, Cytarabine) with E.V.A. (Eltanexor, Venetoclax, A1210477) inhibiting XPO1, Bcl-2 and Mcl-1.” is overall interesting, logical, and presented in an easily understood manner. A few comments below:

1.     Line 406-408: In U87 and U251, the Mcl-1 mRNA expression was increased in response to Venetoclax treatment. Therefore, in this case, it would be hard to say that it is post-translational upregulation. Please clarify this.

2.     In the cell cycle analysis, please include the M phase with the G2 phase (G2/M).

3.     The correct notation would be "GBM Grade IV", instead of °IV or °4

4.     Instead of "CO²", the correct notation is "CO₂"

Reviewer 4 Report

Comments and Suggestions for Authors

To the authors:

Zhao et al. investigated the combinatorial effect of TMZ, MTX, and Ara-C with XPO1, Bcl-2, and Mcl-1 inhibitors, which are Eltanexor, Venetoclax, and A1210477. Based on their data, combining these drugs efficiently induced apoptosis in glioblastoma cell lines and glioblastoma stem-like cells. In addition, the selection of available drugs keeps the neuronal toxicity low. Treatment with the standard procedure, including TMZ, has limitations due to the therapeutical resistance of glioblastoma cells; therefore, alternative inducers of, for instance, apoptosis in glioblastoma cells are urgently needed. The manuscript is well-written, and the data supports the conclusion.  

Minor Concern:

1. Figure I: Referring to Figures I.4 and I.5 is missing in the text; “etc” in the figure legend should be avoided.

2. Method part: Prism software, which version was used?

3. Line 329 should be “cell” instead of “cells”; line 332 is missing “for” when referring to “10 days GSC”; and line 569 constantly “caspase 3” should be used instead of “caspase III”.

4. Line 615: “TMZ causes a malignant transformation by itself”, this should be explained. 

Comments on the Quality of English Language

In general well-written.

Reviewer 5 Report

Comments and Suggestions for Authors

The paper “A Novel Approach for Glioblastoma Treatment by Combining Apoptosis Inducers (TMZ, MTX, Cytarabine) with E.V.A. (Eltanexor, Venetoclax, A1210477) Inhibiting XPO1, Bcl-2 and 4 Mcl-1”  is interesting  and a large amount of experimental work was conducted to support the preliminary hypotheses.

The model systems proposed are stabilized GBM lines, GSCs stem-like cells and healthy mouse brain slices. Glioblastoma multiforme (GBM) is a grade IV glioma, the most common malignant primary brain tumours in adults. Its primarily characteristic is its high malignancy and invasion ability that renders a complete surgical resection impossible. Therefore, current treatment options involve maximal surgical resection of the tumour, followed by chemotherapy and radiotherapy that are aimed at destroying the remaining cells. However, these remaining cells can activate survival pathways and acquire resistance to both chemo and radiation therapy, most patients having tumour recurrence, that are often times more aggressive than the initial tumour. This is why, even with this full treatment scheme, GBM patients have an average survival time between 12- and 15- months following diagnosis. Current research is focusing on reducing the tumoral cells survival, either by targeting specific pathways used by the tumour for survival or by targeting pathways that offer the tumour resistance to chemo and radiation therapy. Resistance to therapeutic treatment is the main reason causing death of cancer patients, especially of glioblastoma. The standard care protocol for glioblastoma is temozolomide (TMZ)-mediated chemotherapy conjugated with radiotherapy. The tumor-suppressive effect of TMZ is restricted within a short window caused by the high frequency of recurrence in glioblastoma. Acquired chemoresistance is a serious limitation to the therapy with more than 90% of recurrent gliomas showing little response to a second line of chemotherapy. Therefore, it is necessary to explore alternative strategies to enhance the sensitivity of glioblastoma (GBM) to TMZ in neuro-oncology.

The authors conducted an interesting study on the combined action of apoptosis inducers like TMX, MTX and Cytarabine with XPO1/Bcl-2/Mcl- 25 1-inhibitors such as Eltanexor, Venetoclax and A1210477 supported by encouraging results. They also tested the response to treatments not only on stabilised GBM lines but also on GSCs stem-like cells that are known to be responsible for GBM chemoradioresistance to therapies. All this is certainly interesting, but they first focused their attention on only one of the possible mechanisms of cell death, i.e. apoptosis, not considering other possible mechanisms of cell death that can be targeted in therapies such as ferroptosis, which is attracting great interest in GBM therapy. By being a non-caspase dependent form of cell death, ferroptosis presents as a promising process that could be involved in GBM treatment. By inducing it in GBM, cancer cells growth and differentiation is inhibited. Inhibiting the xCT system, reducing cysteine levels and thus GSH levels, as well as reducing GPX4 activity and increasing iron availability and ROS formation, each stimulate lipid peroxidation and thus, promote ferroptosis that in turn limits the cells’ ability to survive and to develop mechanisms of resistance to treatments.

 Moreover, the authors themselves at the conclusion of their experiments raise the great bias of their study: systemic application routes, various studies have reported that in humans, the plasma levels of Venetoclax range between 30 and 4000 ng/mL, depending on the dosage administered (from 100 to 600 mg). In the quest to analyze, whether oral Venetoclax administration leads to sufficient doses in the CNS they used CSF as an indicator for distribution. Oral application of Venetoclax, as exemplified for 2 patients, resulted in approximately 1/340th of the serum concentration in cerebrospinal fluid and is hence not efficient to reach therapeutic doses in patients with an intact blood brain barrier.

The crossing of the blood-brain barrier that limits the applicability of this study can be overcome with different approaches such as the use of carriers or molecules capable of crossing it

As a means to improve the efficacy of TMZ and reduce the side effects of chemotherapy, systemic TMZ administration using a biodegradable carrier such as nanoparticles is widely studied. Since the nanoparticles allow targeted drug delivery, the preparation may reduce the administrative dose. Research, to date, has focused on preparing nanoparticles that possess several properties, such as enhanced ability to reach the brain, cross the blood– brain barrier, and bind specifically to receptors that are over-expressed in glioma cells.

Finally small molecules such as Cucurmin, a bioactive compound derived from Indian spice Cucurmin longa (C. longa), is a well-known antioxidant and anticancer agent modulates cell proliferation and apoptosis via a number of signaling pathways, and detailed action of cucurmin was recently reviewed. Of note, cucurmin crossed the blood–brain barrier and reduced the tumor size. The effect of cucurmin was enhanced when it was administered together with a phospholipase A2 alpha inhibitor, which acts by inhibiting the formation of lipid droplets in glioma cells.

However, the final results authors reported on experiments carried out on healthy mouse brain slices where only low side effects in vivo were observed seem to encourage further studies to overcome the BBB.

General comment

The authors are asked to discuss the limitations of their study in more detail with pros and cons in the discussion section, and to provide possible solutions to overcome the limitations found.

Comments on the Quality of English Language

English is fine

Round 2

Reviewer 5 Report

Comments and Suggestions for Authors

The authors made an effort to respond to the referee's requests by giving reasons at each point for the choice of experimental design they followed. They also reported the main limitations of the study's applicability in the discussion session. The work in this version can be accepted for publication in Cells

Comments on the Quality of English Language

English is fine